# Exploring Key Barriers of HACCP Certification Adoption in the Meat Industry: A Decision-Making Trial and Evaluation Laboratory Approach

**DOI:** 10.3390/foods13091303

**Published:** 2024-04-24

**Authors:** Adriana Dima, Elena Radu, Cosmin Dobrin

**Affiliations:** 1Faculty of Management, The Bucharest University of Economic Studies, 010374 Bucharest, Romania; cosmin.dobrin@man.ase.ro; 2Faculty of Business Administration, The Bucharest University of Economic Studies, 010374 Bucharest, Romania; elena.radu2102@gmail.com

**Keywords:** DEMATEL, foodborne hazards, food safety, HACCP, meat industry

## Abstract

Food safety management represents an important concern in contemporary society. The Hazard Analysis Critical Control Point (HACCP) system is a crucial tool for meat producers, preventing and controlling major food safety concerns in the process. This research investigates key barriers to HACCP implementation in the meat industry, employing the Decision-Making Trial and Evaluation Laboratory (DEMATEL) model to identify and categorize these obstacles. Using the insights of 18 experts, a cause-and-effect relationship diagram is generated through which twelve barriers are categorized in terms of their cause and effect and then analyzed. Threshold value is calculated as 0.299. The findings reveal poor management as the primary impediment, followed by challenges in training, knowledge, and dedication. Categorizing the barriers into four groups emphasizes the critical role of effective management and human resources. The study contributes valuable insights to food safety management literature, serving as a practical resource for industry practitioners. Despite limitations in relying on expert opinions and the industry-specific focus, the research lays a foundation for informed decision-making, stressing the importance of effective management in successful HACCP implementation. Future research directions include diversifying geographical representation, exploring practical solutions, and integrating emerging technologies for a better understanding of HACCP adoption challenges.

## 1. Introduction

Food safety management represents an important concern in the modern world. Traditional methods of preventing and controlling foodborne hazards have become increasingly ineffective as a result of the increasing complexity of the food supply chain, globalization, and social changes in people’s lifestyles [1]. The likelihood of large-scale outbreaks of foodborne diseases has increased due to the industrialization, mass production, and distribution of food [2]. In the modern era, a large number of samples would need to be tested to ensure a certain level of safety, which makes testing the finished product insufficient for assurance. In practice, adequate testing of the finished product to obtain reliable information is not economically feasible and the results would often be received after the product had been marketed or consumed. This turning point in the quality control process determined the development of Hazard Analysis Critical Control Point (HACCP), a risk management approach designed to address food safety concerns in the food industry. Over time, HACCP has proven its functionality when integrated with general cleaning rules, rather than being used in isolation. General hygiene codes are used as a first line of defense to ensure that food production, processing, or other processes are sanitary in general (transport, distribution, preparation, etc.). As a second line of defense, HACCP is used to create a customized food safety assurance system for the product in question; testing is used as the final line of defense to ensure that preventive measures are working [3]. The adoption of HACCP principles in the operational activities of food businesses has become an essential component in the European Union through the General Food Law Regulation (EC) No 178/2002, reflecting its recognition as a key strategy for managing food risks [4]. Implementing national and international regulations helps organizations to significantly enhance their sustainability in the market [5].

In spite of the numerous advantages of HACCP implementation, its adoption faces complex challenges influenced by managerial, organizational, and technical factors [6]. Understanding the causes that constitute barriers to HACCP certification adoption in the food industry is an important issue for companies in the industry and a starting point for future research in this field.

The current research is guided by the following objectives:(O1)Identifying the critical barriers that impede the successful adoption of HACCP certification within the meat industry.(O2)Creating a thorough causal relationship model that reveals the interdependencies between the identified barriers.(O3)Categorizing the barriers into distinct cause-and-effect groups to assess their relative significance within the context of HACCP certification adoption.

To achieve the abovementioned objectives, this research employs the Decision-Making Trial and Evaluation Laboratory (DEMATEL) methodology. In complex decision-making with multiple criteria, using methods such as DEMATEL from the field of Multiple Criteria Decision-Making (MCDM) represents a decisive factor. DEMATEL is a structural modeling method applied to analyze cause-and-effect relationships, confirm interdependencies, and evaluate their strengths [7]. It facilitates capturing group knowledge, creating a structural evaluation model, and visualizing causal relationships among subsystems. Based on directed graphs, DEMATEL separates criteria into cause-and-effect groups, facilitating the understanding of system dynamics and emphasizing influential criteria within the causal network [8].

This research initially conducts a rigorous literature review to identify significant barriers associated with the implementation of HACCP. Based on the main findings, expert questionnaires are conducted to collect insights from experienced practitioners within the meat industry. The results are analyzed using the DEMATEL method to demonstrate cause-and-effect relationships underlying these barriers.

The novelty of this research resides in the fact that no prior studies have applied the DEMATEL method to investigate barriers in HACCP adoption within the meat industry. Moreover, to the authors’ knowledge, there is no research that addresses the barriers of HACCP application in the Romanian meat industry. The study investigates an ongoing and significant concern in the meat industry, specifically the main barriers of HACCP implementation, being the first approach of the DEMATEL method in this context. Using DEMATEL to create a detailed model of the cause–effect relationships between these barriers facilitates the understanding of system dynamics and influential criteria in the HACCP certification process by industry organizations, researchers, and interested policy-makers.

## 2. Literature Review

HACCP represents a systematic and operation-specific risk management approach to ensure food safety [9], evolved globally as the primary method for securing the food supply. HACCP’s systematic approach plays a pivotal role in preventing microbiological contamination, and enhancing food quality and certification under HACCP is perceived as a crucial tool for producers to signal product quality and safety to consumers [10]. HACCP’s importance resides in its capacity to prevent and control food safety risks, providing a comprehensive framework for food safety management in various sectors of the food industry [11,12].

Incorporating all barriers for HACCP implementation in the meat industry into the present analysis is not practically viable. The study initially derived a list of barriers through a literature review, as summarized in Appendix A. To reduce the number of barriers identified and to include those that are current and suitable for the study, the initial set was reviewed by eight experts from the food industry sector and six experts from the academic world. The expert group narrowed down the number of barriers from 36 to 12 (referred to as factors), which are included in the DEMATEL questionnaire.

To identify the initial set of barriers, a systematic literature review was undertaken. A total of 68 publications were identified from the Web of Science Core Collection for “barriers” (Abstract) and “HACCP” (Abstract) in the timeframe 1994–2023. In an editorial piece from 1994, Kirby stated that there are challenges related to technical resources and the concentration of functions, time, and financial resources to HACCP application in SMEs [13]. Subsequent studies up to the year 2000 focused on barriers to international trade but identified human factors as the main barriers to HACCP implementation: lack of knowledge, expertise, sufficient resources [14], company culture [15], communication [15], time [15], and motivational barriers aimed at shifting employees’ views of themselves and facilitating behavior change [16].

Since 2001, studies addressing these concepts have increased in intensity but also in importance for scholars, receiving a significant number of citations. Panisello and Quantick [17] provided a broad theoretical classification of HACCP barriers according to the implementation step into barriers that appear before HACCP implementation (managers’ illusion of control, company size, product type, industry field, and organizations’ safety requirements), during the process (management, staff, and infrastructure), and after HACCP implementation (verification and evaluation difficulties, and lack of equivalency between related programs). Gilling et al. [9] first investigated complex behavioral barriers to following HACCP guidelines, identifying 11 barriers organized on three levels: behavior (external/customer, guidelines, environmental, competence, and cueing mechanism factors), attitude (motivation, outcome expectancy, self-efficacy, and agreement), and knowledge (understanding and awareness). Following this model, Vela and Fernández [18] applied a survey in Madrid. The findings indicated that inadequate hazard analysis is caused by negative guidelines and a lack of comprehension—a problem that cannot be resolved by consulting outside experts. Additionally, they identified issues at the attitude level that prevent behavior modification [18]. Based on the same model, Taylor and Taylor [19] conducted a qualitative psychology study to investigate the interplay of HACCP successful implementation in four organizations from North West England. Five major “barriers” to successful implementation were identified: HACCP was considered challenging, onerous, and needless, impeded by personnel and external issues [19]. Jevšnik et al. [20] used a systematic review to identify barriers from previous studies and clustered them into 21 new categories.

Baş et al. [6] investigated the challenges to HACCP adoption and food safety systems in Turkish food enterprises, based on a questionnaire prepared on the previous survey conducted by the Food Safety Authority of Ireland. The results suggested that the main barriers were a lack of understanding of HACCP, lack of prerequisite programs, lack of knowledge about HACCP, lack of time, turnover, lack of employee motivation, complicated terminology, and lack of personnel training [6]. Based on 20 potential barriers identified from the literature review, Herath and Henson [21] applied a questionnaire in Ontario, Canada and concluded that financial constraints were the most significant barrier. Other common HACCP barriers identified by other research include information transmission [22], lack of training [23], time [23,24], non-awareness to HACCP guidelines [25], and financial constraints [23,26].

Since 2011, studies in this field have mainly focused on practical research, applying and reviewing the concepts introduced in the previous period. Most of the practical studies involve interviews or questionnaires. Using face-to-face interviews and questionnaire methods applied for 28 dairy plants from the Aydin region in Turkey, Karaman et al. [27] found that the main barriers to HACCP and other food safety programs in the dairy industry are lack of knowledge and high costs [27]. Based on four narrative interviews, Lowe and Taylor [28] investigated barriers to HACCP implementation among farmers and fresh produce growers from England, who consider HACCP difficult, costly, and unnecessary, acting as barriers to implementation [28].

The study of Al-Busaidi et al. [29] showed that firms in fish processing in the Sultanate of Oman, Turkey face cost, and lack of staff expertise and commitment, as HACCP implementation barriers [29]. Previous research on Turkish seafood processing companies found inadequate training for staff members as the main HACCP implementation barrier [30]. Based on a cross-sectional qualitative research design, a study in milk processing companies in Armenia indicated that the main barriers to HACCP Food Safety Management System adoption include high costs, value incompatibility, excessive documentation, and inadequate physical and technological infrastructure [31]. A more recent study investigated the practical barriers to HACCP implementation on dairy farms in South Korea. The results pointed to a lack of adequate financial assistance, incentive programs, HACCP consultation and education, consumer awareness and knowledge about HACCP certified products, effective record-keeping systems, and farm-oriented mobile operation and maintenance systems [32].

In terms of bibliometric studies and systematic reviews, there have been a number of important studies. Fotopoulos et al. [33] identified 32 factors influencing HACCP implementation in the global food processing industry after reviewing articles published between 1995 and 2008. Milios et al. [34] outlined a lack of awareness, perceived benefits, and training as well as ineffective management; variability of production lines and products; variable consumer requirements and small business size; and costs of system development, implementation, and maintenance as obstacles to implementing a HACCP system [34]. In 2019, Yang et al., based on a meta-analysis technique, indicated that many Chinese food producers, especially small ones, face difficulties and barriers such as financial constraints, lack of prerequisite programs, and frequent staff turnover to implementing HACCP-based food safety management systems [35]. A more recent bibliometric study grouped barriers for implementing HACCP into categories and the most common barriers refer to budgetary constraints, knowledge and perception of employees, and bureaucracy [36].

The revised list of HACCP implementation barriers in the meat industry is presented in Table 1 below.

## 3. Materials and Methods

DEMATEL and other variants have predominantly been employed in limited studies related to food. A study from 2018 applied Grey DEMATEL to examine factors influencing entrepreneurship [37]. Based on the opinion of 13 experts from Iran, environmental factors have the greatest influence and individual factors are the most influenced and crucial in the entrepreneurship process within the food industry [37]. Khan et al. [38] identified ten significant obstacles to the implementation of halal certification through assessment and accreditation (HCAA) in India with the help of five experts and used fuzzy DEMATEL to examine the connection between cause and effect among the barriers. “Halal” describes goods suitable for Muslims, but typically, the hygiene and security of such products are disregarded. Findings highlight that top management commitment and government sup-port have the greatest influence, while lax enforcement and halal logo compliance represent the biggest obstacles. Strong management commitment and government assistance are considered to be of high importance [38]. Prakash et al. [39] identified critical success factors (CSFs) for the ice-cream industry using DEMATEL methodology. Results indicate that infrastructure and capacity building, consistent product improvement, and operational efficiencies constitute key factors to industry growth and the use of IT and improved operational processes significantly impact the industry’s performance [39]. Kumar et al. [40] investigated the barriers of Industry 4.0 (I4.0) adoption applications for a circular economy in sustainable food supply chains (SFSCs) using the Rough-DEMATEL technique. Findings indicate that technological immaturity, high investment, lack of awareness and customer acceptance, and technological limitations and lack of eco-innovation as the main barriers to I4.0 adoption in SFSCs.

An in-depth scientific review highlighted critical barriers to HACCP certification adoption in the food industry and the DEMATEL technique was used to determine the causal interrelationships among these elements. The DEMATEL methodology represents an effective causal analysis technique that categorizes key barrier factors into cause and consequence groups [41] and enables the development of a visual representation that demonstrates the cause-and-effect interaction between these factors. The approach is built on a two-phase procedural framework [42].

Phase 1 resides in rigorous research methodology in order to identify all previously mentioned barriers in scientific articles.

Phase 2 implies using the DEMATEL method to investigate and provide a systematic representation of the causal connections between the selected factors. The sample for the final DEMATEL questionnaire includes a total of 18 interviewees: 14 experienced meat industry experts and practitioners, and four academics researching the food industry. The DEMATEL methodology transforms qualitative information into quantitative analysis [43], and the number of required reference experts is not subject to any specified criteria [37]. In previous research, this starts from six experts [38]; so, it can be stated that in this research the sample is sufficient for the analysis undertaken. Phase 2 can be developed through the following three steps.

*Step* *1:*
*Generating the Direct-Relation Matrix (M).*


On a scale from 0 to 4 (Table 2), each of the 18 experts in the sample was asked to conduct pairwise comparisons between the 12 factors. Equation (1) underlies the initial-relation matrix that indicates the experts’ individual opinions and evaluations on the causality between the factors [42,44,45]. Equation (1) shows there are e experts involved, where e = {1, 2, 3 … x}. $M_{e}$ evaluates each expert’s interaction choices among the factors. For this research, a total of 18 matrices were generated, each consisting of 12 rows and 12 columns.
(1)$$M_{e}=\left[\begin{array}{lllll} 0 & m12 & m13 & \ldots & m1n\\ m21 & 0 & m23 & \ldots & m2n\\ m31 & m32 & 0 & \ldots & m3n\\ \ldots & \ldots & \ldots & \ldots & \ldots \\ \ldots & \ldots & \ldots & \ldots & \ldots \\ mn1 & mn2 & mn3 & \ldots & 0 \end{array}\right]$$

*Step* *2:*
*Computing the Standardized Direct-Relationship Matrix (X)*


The direct-relation matrix (X) is calculated by applying the formula in Equation (2) [42,44,45]:
(2)X=k×M
(3)$$k=\frac{1}{\underset{1\leq i\leq n}{\max}\sum_{j=1}^{n}m_{ij}},\;i,\;j=1,\;2,\;\ldots \;n$$

*M* is the initial-relation matrix given by Equation (1); *k* is the average of all expert $m_{ij}$ values, calculated according to Equation (3); and *X* is the normalized direct-relation matrix. The values in each column of the normalized direct-relation matrix must be less than one for the DEMATEL technique to be relevant [42,45,46].

*Step* *3:*
*Calculating the Total-Relation Matrix (T)*


The total-relation matrix (T) is computed through Equation (4) [42,44,45]:(4)$${T=X(I-X)}^{-1}$$

*I* represents the identity matrix, *T* the total-relation matrix, and *X* the direct-relation matrix specified in Equation (2). The total-relation matrix *T* helps calculating the sum of the number of rows (D) and columns (R).

*Step* *4:*
*Determining the Causal Parameters*


Equations (5) and (6) are used to compute the causal parameters (D and R) within the T matrix [42,45,47].
(5)$$(D)={\left[d_{ij}\right]}_{n\times 1}={\left[\sum_{j=1}^{n}d_{ij}\right]}_{n\times 1}$$
(6)$$(R)={\left[r_{ij}\right]}_{1\times n}={\left[\sum_{j=1}^{n}r_{ij}\right]}_{1\times n}$$

*Step* *5:*
*Determining the Prominence and Effect Score*


The sum of all elements in the averages in matrix T is calculated and divided by the total number of elements in the matrix to obtain the threshold value (∝). ∝ is obtained using Equation (7) [45,48]:(7)$$\propto =\frac{\sum_{j=1}^{n}\sum_{i=1}^{n}r_{ij}}{n^{2}}$$

$n^{2}$ represents the total number of entries in the total relation matrix T. Considering that there are *n* barriers, the total number of elements in matrix is T = *n* × *n* = $n^{2}$ [43,47]. Following that, a connection diagram is created by charting the values of (D + R) and (R − C). The Y-axis represents the values of (D − R), and the X-axis represents the values of (D + R). A directed graph illustrates the interdependence of the main factors. Values in the T matrix that correspond to or exceed α are thought to have a considerable effect. The directed graph is created using the influential strength matrix [42]. The results of D and R confirm the level of relational influence among each critical factor.

## 4. Results and Discussion

The first section of the DEMATEL questionnaire includes demographic items, which allow the analysis of the sample of respondents. A summary of the results for this section is presented in Table 3. The distribution of respondents across different industry types, organization sizes, positions, industry experiences, and professional qualification levels contributes to the comprehensiveness of the study, ensuring diverse perspectives and expertise in evaluating HACCP implementation barriers. In the meat production sector, respondents from organizations with 51–100 employees hold positions as Owner/Proprietor (2), Senior Manager (2), and Food Safety Specialist (2), showcasing a balanced representation across various roles. Another group from organizations with 101–500 employees includes Owner/Proprietor (1), Senior Manager (2), and Food Safety Specialist (4), suggesting a broader range of expertise, especially among Food Safety Specialists. The presence in the sample of researchers with Ph.D. qualification from organizations with over 500 members (4) indicates a high level of academic expertise, emphasizing the significance of research experience and advanced qualifications.

All organizations within the meat production industry included in the study have a minimum operational history of 10 years and produce a minimum of three distinct products. Given the variability in meat types and production processes, quantifying volume in tons per year was deemed impractical and irrelevant for meaningful comparison. It is pertinent to note that all companies in the sample maintain internal personnel dedicated to HACCP plan oversight, with no external outsourcing for this function. Respondents’ professional qualification levels correspond to their nationally recognized educational diplomas, although the majority reported participation in short-term training programs focused specifically on food safety.

The experts whose demographic profile was presented above had to conduct pairwise comparisons between the 12 factors. Based on the centralization of each set of responses received, the average matrix M was calculated (Appendix B).

Further on, the direct-relation matrix (X) was calculated based on the initial-relation matrix (average matrix) and the normalized direct-relation matrix (Table 4). As it can be noted, all the values in the normalized direct-relation matrix are less than one, indicating that the DEMATEL technique is relevant for this research.

Following this, as indicated in Equation (4), the normalized initial direct-relation matrix was used to compute the total relation matrix (T). The threshold value (∝) was calculated to identify the relationships between barriers and to develop the cause-and-effect diagram. It resulted that ∝ is 0.299, and Table 5 presents the total relation matrix (T); highlighted are the barrier values equal or higher than ∝.

Next, the casual parameters are calculated (D, R values) using Equations (5) and (6), as presented in Table 6. Table 6 shows the obtained values, the identity of each factor included in the analysis, and the ranking of Di + Rj values. The Di + Rj value presents the importance of the determinants while the Di − Rj value identifies the nature of the factor: cause or effect.

Based on Di + Rj ranking, the importance order of the 12 factors is as follows: Ineffective management (F3) > Lack of or inadequate personnel training (F12) > Lack of staff expertise and commitment (F4) > Lack of knowledge about HACCP (F7) > Verification and evaluation difficulties (F5) > Company size (F1) > Financial constraints (F6) > Product type (F2) > Time (F8) > Complicated guidelines (F10) > Volume of paperwork (F11) > Lack of physical conditions (F9).

Upon carefully analyzing the research findings, it becomes apparent that the identified barriers to HACCP implementation in the meat industry are deeply intertwined with organizational dynamics and operational complexities. In particular, the cause-and-effect analysis reveals intricate relationships among various factors, shedding light on the underlying mechanisms influencing the success or failure of HACCP initiatives within meat processing facilities.

Positioned as the most critical barrier, ineffective management (F3) demands immediate attention and intervention, highlighting the important role of leadership and decision-making in ensuring the success of HACCP systems. This finding resonates with the existing literature, which consistently underscores the significance of managerial commitment and support in driving food safety initiatives within organizations [6,17,34]. It is essential to recognize that managerial deficiencies often stem from broader organizational issues, such as a lack of prioritization of food safety [21], inadequate resource allocation [6,13,21,23,26,27,28,29,31,32,35], or resistance to change [15]. Therefore, addressing managerial shortcomings necessitates a holistic approach that involves fostering a culture of food safety, promoting transparent communication channels, and empowering leaders to prioritize HACCP compliance throughout the organization [15].

Lack of or inadequate personnel training (F12) is the second most important determinant, emphasizing the pivotal role of well-trained personnel in effective HACCP implementation. This not only reflects deficiencies in individual competencies but also highlights systemic issues related to workforce development and capacity-building. This finding aligns with prior research highlighting the importance of employee education and skill development in ensuring compliance with food safety standards [6,23,30,34]. To address this barrier effectively, organizations must invest in comprehensive training programs tailored to the specific needs of their employees, ensuring that staff possess the requisite knowledge and skills to execute HACCP protocols effectively [30].

Ranking third, lack of staff expertise and commitment (F4) represents another important barrier related to human resources. This emphasizes the need for a knowledgeable and dedicated workforce [6,35] to overcome implementation challenges. The proficiency and qualifications of personnel overseeing food safety and quality are vital for effective HACCP system implementation, particularly in animal-derived food sectors. These individuals require a profound comprehension of slaughter procedures; biological and chemical hazards associated with livestock, including zoonotic risks; and meat microbiology. It is noteworthy that if the professional qualifications of food safety and quality personnel align with conventional educational diplomas, such as bachelor’s or master’s degrees in food engineering or business, these professionals may exhibit deficiencies in specialized knowledge crucial for organizational efficacy.

The 12 factors were divided into two groups based on (Di − Rj) values: cause group and effect group (Table 6). The factors having Di − Rj > 0 are classified as a cause barrier and influence the others directly. The factors with Di − Rj < 0 are categorized in the effect category and are influenced primarily by some of the ones in the cause group. A total of six factors—namely, Company size (F1), Time (F8), Lack of physical conditions (F9), Complicated guidelines (F10), Volume of paperwork (F11), and Lack of or inadequate personnel training (F12)—are categorized as cause barriers. The remaining six factors—namely, Product type (F2), Ineffective management (F3), Lack of staff expertise and commitment (F4), Verification and evaluation difficulties (F5), Financial constraints (F6), and Lack of knowledge about HACCP (F7)—are effect barriers. In the next cause–effect diagram (Figure 1), the “cause barriers” are shown on the positive side of the Y-axis while the “effect barriers” are on the negative Y-axis.

F12 is identified as the most important cause in the implementation of HACCP in the meat industry. This indicates that investment in adequate staff training could be essential to improve the effectiveness and success of HACCP implementation in such organizations. Also, as can be seen from the Total Rotation Matrix Table, the influence of F12 is present on all factors included in the analysis except F9 and F10, which shows the significant impact that a lack of or inadequate personnel training has on many other parts of the organization. The need for adequate staff training is evident, suggesting that strategies focused on developing employees’ skills and knowledge can significantly contribute to overcoming the identified barriers. A short discussion is needed for F10: complicated guidelines. Sectorial guidelines are criticized for their lack of specific and actionable recommendations, making it challenging for food businesses to implement HACCP effectively. Official guidelines, while more comprehensive, are often inaccessible or difficult to interpret due to complex language and technical terminology. Moreover, inconsistencies between different sources of information contribute to confusion among stakeholders, hindering efforts to ensure food safety compliance [49].

The most important effect is F3, signifying its vulnerability to and influence on other barriers. This finding underlines the importance of effective management in food organizations to ensure the success and effectiveness of the HACCP system. It is evident that approaches focused on improving management qualities, as well as the implementation of effective management practices, could significantly contribute to overcoming this barrier and improving overall HACCP implementation.

While factors like company size (F1) and lack of physical conditions (F9) are categorized as cause barriers, their impact is further compounded by external pressures, such as financial constraints (F6) and complicated regulatory guidelines (F10). This interdependence underscores the need for a holistic approach to HACCP implementation that addresses both internal operational challenges and external regulatory pressures [13,23].

Addressing the cause barriers, such as company size and lack of physical conditions, requires tailored approaches to attenuate their impact on HACCP implementation. For instance, smaller food businesses may face resource limitations and organizational constraints, necessitating targeted support mechanisms from regulatory bodies or industry associations. Collaborative initiatives that provide financial assistance, technical guidance, and access to shared resources can help smaller enterprises overcome these challenges and implement robust HACCP systems effectively. Similarly, investments in infrastructure upgrades and facility improvements can enhance physical conditions, ensuring compliance with food safety standards and facilitating smoother HACCP implementation processes across the industry.

Focusing on effect barriers, such as ineffective management and verification difficulties, demands proactive measures to enhance organizational capabilities and streamline operational processes. Food industry leaders must prioritize leadership development initiatives aimed at developing strong managerial skills and fostering a culture of continuous improvement. By empowering managers with the knowledge and skills necessary to navigate complex regulatory landscapes and drive organizational change, businesses can overcome barriers associated with ineffective management and establish a foundation for successful HACCP implementation. Additionally, investing in technology-driven solutions for verification and evaluation, such as automated monitoring systems and digital record-keeping platforms, can streamline compliance processes, reduce administrative burdens, and improve data accuracy and transparency. Integrating these technological advancements into existing quality management systems can improve the efficiency and effectiveness of HACCP implementation efforts, allowing food businesses to achieve and maintain high food safety and regulatory compliance standards.

Analyzing the results, depending on their rank, cause and effect factors alike can be grouped into four categories: management and human resources, assessment and documentation, resources and organizational structure, and product and time-related aspects (Table 7). The rank indicates their relative importance within each category, providing insights into prioritizing interventions for more effective HACCP compliance.

The most important category is management and human resources, which comprises the top three most important factors included in the analysis. These findings emphasize the importance of effective management and human resources in successful HACCP implementation in the meat industry. Leadership and decision-making play a critical role, and continuous training programs are necessary to enhance staff capabilities. Assessment procedures and documentation also need to be streamlined and well-organized to ensure effective implementation. Organizational size, financial constraints, and physical conditions can pose challenges, and tailored approaches are needed for different product types. Time-related factors also influence HACCP success, highlighting the need for timely processes and interventions. Overall, the results highlight the centrality of organizational and human factors in the successful adoption of HACCP.

Based on Table 6, the net cause/effect graph was generated, as illustrated in Figure 1. Comparing the elements with the ∝ value, the DEMATEL map for the HACCP implementation barriers is developed to present the relationship between the factors (Figure 2). The arrow-headed lines in Figure 2 represent the causal interactions from one factor to another.

Factors in the “Effect group” (F2, F3, F4, F5, F6, and F7) are influenced by the ones in the “Cause group” (F1, F8, F9, F10, F11, and F12), which impedes the implementation of HACCP in the meat sector. F3 and F12 are the most influenced barriers, with all others impacted, followed by F4 (influenced by all except F2) and F7 (influenced by F1, F3, F4, F5, F6, F8, F10, and F12). In turn, F12 influences all of the barriers except F9 and F10. F2 has the least influence on the other factors, impacting only F3 and F12.

This categorization facilitates targeted interventions by allowing stakeholders to prioritize efforts based on the relative importance of different factors within each category. For example, interventions aimed at improving managerial effectiveness and enhancing staff training may yield substantial benefits in addressing the most critical barriers identified in the management [6,14,34] and human resources category [14,29].

## 5. Conclusions

The current research underscores the evolution of food safety challenges amid the complexities of the modern food supply chain. Recognizing HACCP’s global significance, the research emphasizes its role in preventive risk management and quality assurance. Despite HACCP’s acknowledged advantages, this study tackles a major industry issue, examining the main barriers to HACCP implementation in the meat business. To identify barriers, the research reviews the vast literature and surveys 18 experts. DEMATEL is used to study cause-and-effect interactions behind these barriers, contributing novelty by applying this approach to the unique context of the Romanian meat industry.

The threshold value was calculated as 0.299, and the findings reveal that the biggest obstacle in HACCP implementation is poor management, followed by poor training, knowledge, and dedication. Lack of HACCP expertise, verification and evaluation issues, company size, financial constraints, product typology, time, complicated guidelines, paperwork, and physical conditions are other impediments.

This is the first DEMATEL research to identify and group the main barriers to HACCP implementation in the meat industry into four categories: management and human resources, assessment and documentation, resources and organizational structure, and product and time-related aspect. The results find management and human resources to be the most important category, including ineffective management, lack of or inadequate personnel training, and lack of staff expertise and commitment as the main factors.

These findings underscore the pivotal role of human resources in successful HACCP compliance, emphasizing the need for robust management practices and well-trained personnel. Notably, inadequate personnel training ranks as the most significant cause barrier, indicating its pervasive influence on other aspects of organizational functioning. Personnel overseeing food safety require expertise in slaughter processes, biological hazards, and meat microbiology for effective HACCP implementation in animal-derived food sectors. However, many staff holding bachelor’s or master’s degrees may lack specialized knowledge in these areas.

The study underscores the importance of addressing human resource constraints and improving management practices to enhance HACCP implementation effectiveness. Furthermore, the findings emphasize the centrality of organizational and human factors in successful HACCP adoption, with tailored approaches needed to address challenges related to management, assessment, resource allocation, and product-specific considerations.

The research contributes valuable knowledge to food safety management, filling a gap in literature and serving as a resource for industry practitioners, researchers, and policymakers. Also, the findings provide a foundation for informed decision-making, emphasizing the importance of effective management and human resources in successful HACCP implementation.

While this study offers valuable insights into the key barriers of HACCP certification adoption in the meat industry, several limitations should be acknowledged. Firstly, the research predominantly relies on expert opinions, potentially introducing subjectivity. Additionally, the focus on the Romanian meat industry may limit the generalizability of results to other geographical regions or industries. Future research could increase geographical representation and industry inclusivity to enhance the external validity of the findings. Future studies could explore practical solutions and specific strategies to overcome the barriers and evaluate their effectiveness. Finally, given the evolving landscape of food safety and technological advancements, future studies should consider the integration of emerging technologies and evolving regulatory frameworks in their analyses to provide a more nuanced understanding of HACCP adoption challenges.

## Figures and Tables

**Figure 1 foods-13-01303-f001:**
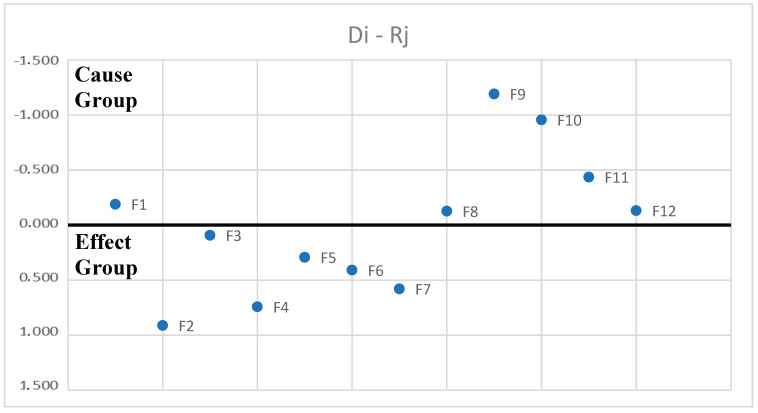
The net cause/effect graph.

**Figure 2 foods-13-01303-f002:**
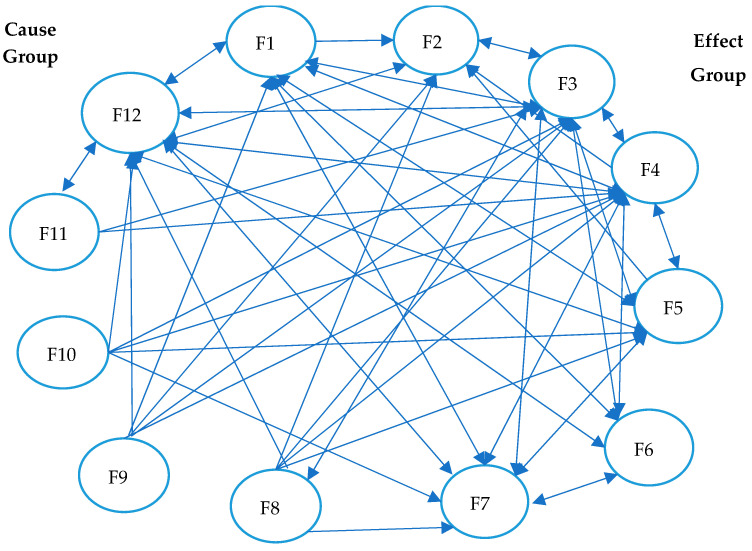
Causal interactions between factors.

**Table 1 foods-13-01303-t001:** Barriers used for DEMATEL.

Code	Barrier/ Factor	References																
		1	2	3	4	5	6	7	8	9	10	11	12	13	14	15	16	17
F1	Company size	x	x															
F2	Product type	x	x	x														
F3	Ineffective management	x	x		x													
F4	Lack of staff expertise and commitment	x				x												
F5	Verification and evaluation difficulties	x																
F6	Financial constraints			x	x	x	x	x	x	x	x				x			
F7	Lack of knowledge about HACCP	x		x								x					x	
F8	Time				x		x	x					x	x				
F9	Lack of physical conditions	x			x		x								x			
F10	Complicated guidelines			x	x							x				x		
F11	Volume of paperwork				x										x			
F12	Lack of or inadequate personnel training		x		x			x										x

Note: 1. [14]; 2. [34]; 3. [32]; 4. [6]; 5. [29]; 6. [13]; 7. [23]; 8. [21]; 9. [28]; 10. [35]; 11. [9]; 12. [15]; 13. [24]; 14. [31]; 15. [18]; 16. [27]; 17. [30].

**Table 2 foods-13-01303-t002:** The DEMATEL scale equivalent value.

Descriptors	Scale
No influence	0
Very low influence	1
Low influence	2
High influence	3
Very high influence	4

**Table 3 foods-13-01303-t003:** Sample of participating experts in the DEMATEL study.

No.	Type of Industry or Field	Organization Size	Company Position	Industry Experience	Professional Qualification Level
1.	Meat Production	51–100 employees	Owner/Proprietor (2)	16–20 Years	Bachelor’s degree
			Senior Manager (2)	11–15 Years	Master’s degree
			Food Safety Specialist (2)	11–15 Years	Professional certification
		101–500 employees	Owner/Proprietor (1)	More than 20 Years	High school diploma or equivalent
			Owner/Proprietor (1)	More than 20 Years	Bachelor’s degree
			Senior Manager (2)	16–20 Years	Master’s degree
			Food Safety Specialist (4)	11–15 Years	Professional certification
2.	Academia/ Research Institution	More than 500 employees	Researcher (1)	11–15 Years	Ph.D.
			Researcher (3)	More than 20 Years	Ph.D.

Note: The figures in () represent the number of respondents in the respective category.

**Table 4 foods-13-01303-t004:** Normalized initial direct-relation matrix (X).

	F1	F2	F3	F4	F5	F6	F7	F8	F9	F10	F11	F12
F1	0	0.124	0.103	0.124	0.029	0.093	0.057	0.029	0.100	0.057	0.029	0.108
F2	0.003	0	0.062	0.019	0.053	0.093	0.067	0.065	0.074	0.038	0.067	0.074
F3	0.117	0.074	0	0.117	0.112	0.103	0.107	0.081	0.060	0.059	0.048	0.105
F4	0.084	0.062	0.091	0	0.102	0.081	0.124	0.074	0.002	0.043	0.040	0.077
F5	0.093	0.076	0.079	0.096	0	0.050	0.089	0.067	0.002	0.052	0.084	0.112
F6	0.098	0.071	0.081	0.076	0.036	0	0.100	0.041	0.028	0.000	0.024	0.100
F7	0.050	0.041	0.096	0.108	0.117	0.062	0	0.045	0.065	0.048	0.048	0.091
F8	0.021	0.088	0.121	0.078	0.108	0.008	0.080	0	0.056	0.030	0.028	0.070
F9	0.124	0.098	0.086	0.067	0.040	0.069	0.026	0.072	0	0.002	0.053	0.079
F10	0.067	0.059	0.110	0.072	0.095	0.010	0.107	0.034	0.021	0	0.059	0.076
F11	0.088	0.050	0.064	0.059	0.048	0.043	0.033	0.095	0.021	0.088	0	0.072
F12	0.077	0.081	0.124	0.124	0.114	0.117	0.102	0.072	0.022	0.083	0.083	0

**Table 5 foods-13-01303-t005:** Total rotation matrix (T).

	F1	F2	F3	F4	F5	F6	F7	F8	F9	F10	F11	F12
F1	0.273	0.386	0.427	0.43	0.322	0.352	0.363	0.258	0.25	0.228	0.222	0.419
F2	0.204	0.196	0.299	0.25	0.262	0.272	0.283	0.227	0.181	0.163	0.203	0.301
F3	0.418	0.382	0.384	0.476	0.439	0.393	0.451	0.335	0.235	0.258	0.267	0.466
F4	0.328	0.31	0.395	0.301	0.37	0.316	0.401	0.279	0.15	0.207	0.218	0.373
F5	0.341	0.329	0.392	0.395	0.283	0.296	0.377	0.28	0.153	0.222	0.262	0.408
F6	0.306	0.285	0.342	0.33	0.272	0.214	0.339	0.221	0.156	0.145	0.178	0.350
F7	0.303	0.291	0.397	0.397	0.381	0.3	0.287	0.256	0.201	0.21	0.226	0.382
F8	0.248	0.306	0.386	0.339	0.348	0.228	0.331	0.192	0.182	0.178	0.192	0.333
F9	0.343	0.326	0.362	0.335	0.284	0.29	0.285	0.261	0.138	0.154	0.213	0.347
F10	0.297	0.288	0.388	0.346	0.345	0.237	0.364	0.231	0.156	0.157	0.225	0.349
F11	0.293	0.263	0.325	0.308	0.279	0.243	0.276	0.266	0.145	0.223	0.151	0.320
F12	0.387	0.388	0.498	0.484	0.446	0.406	0.454	0.331	0.201	0.283	0.300	0.374

Threshold (∝) value = 0.299.

**Table 6 foods-13-01303-t006:** Identity factors.

Factor	Di	Rj	Di + Rj	Rank	Di − Rj	Identity
Company size (F1)	3.931	3.742	7.673	6	0.189	Cause
Product type (F2)	2.84	3.752	6.592	8	−0.912	Effect
Ineffective management (F3)	4.501	4.594	9.095	1	−0.093	Effect
Lack of staff expertise and commitment (F4)	3.65	4.392	8.042	3	−0.742	Effect
Verification and evaluation difficulties (F5)	3.738	4.031	7.769	5	−0.293	Effect
Financial constraints (F6)	3.138	3.547	6.685	7	−0.409	Effect
Lack of knowledge about HACCP (F7)	3.631	4.211	7.842	4	−0.580	Effect
Time (F8)	3.263	3.137	6.400	9	0.126	Cause
Lack of physical conditions (F9)	3.339	2.148	5.487	12	1.191	Cause
Complicated guidelines (F10)	3.383	2.427	5.810	10	0.956	Cause
Volume of paperwork (F11)	3.092	2.657	5.749	11	0.435	Cause
Lack of or inadequate personnel training (F12)	4.552	4.421	8.973	2	0.131	Cause

**Table 7 foods-13-01303-t007:** Main categories of cause-and-effect factors in HACCP implementation barriers.

Category	Factor	Rank
Management and Human Resources	Ineffective management (F3)	1
	Lack of or inadequate personnel training (F12)	2
	Lack of staff expertise and commitment (F4)	3
Assessment and Documentation	Lack of knowledge about HACCP (F7)	4
	Verification and evaluation difficulties (F5)	5
	Complicated guidelines (F10)	10
	Volume of paperwork (F11)	11
Resources and Organizational Structure	Company size (F1)	6
	Financial constraints (F6)	7
	Lack of physical conditions (F9)	12
Product and Time-Related Aspects	Product type (F2)	8
	Time (F8)	9

## Data Availability

The original contributions presented in the study are included in the article, further inquiries can be directed to the corresponding author.

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
