# Peer review of "Exploring Key Barriers of HACCP Certification Adoption in the Meat Industry: A Decision-Making Trial and Evaluation Laboratory Approach"

_foods, 2024, doi:10.3390/foods13091303_

Round 1
Reviewer 1 Report
Comments and Suggestions for Authors
General comments
The presented article utilizes the DEMATEL technique to assess the main problems in the implementation of an HACCP plan in the food industry. The results are consistent with the reality of the companies. The primary issue with the methodology is the limited information collected about the companies. I consider it important to include the volume (in tons) per year, as well as the number of products they produce. It is also not indicated whether all companies have one (or several) individuals dedicated to ensuring the HACCP plan or if it is an external company (outsourcing) that carries it out. Furthermore, it is not specified whether the training is specific to food safety, only indicating whether it is professional, university, or doctoral education.
Lastly (and associated with the scant description of both the characteristics of the sample individuals and the company's characterization), the discussion is scarce or almost non-existent. I find the work interesting but it does not discuss the reasons behind the results.
Specific comments
1. Introduction
1.1. Line 55: HACCP's systematic approach, particularly in primary production, plays a pivotal role in preventing microbiological contamination and enhancing food quality and certification under HACCP is perceived as a crucial tool for producers to signal product quality and safety to consumers.
The HACCP in the primary production is not compulsory in the EU. If authors observe the 7 points of the haccp implementation, the 4th point indicates the “need of monitoring procedures”. For example, in a dairy farm, you can identify Brucella, Mycobacterium spp., Coxiella spp., Chlamydia spp. etc as a microbiological risk. After that, you can monitorize the microbiological counts in the bulk tank (independently its economic viability). But then, how do you apply corrective measures? Now, imagine you produce lettuces, How do authors apply a “monitoring system”? After that explanation, the HACCP does not play a pivotal role in the microbiological contamination in the primary production as stated. It´s impossoble to implement a HACCP program in the primary production.
2. Metodology – results
2.1. In the results sections, author stated that the first 4 factors (line 320-325) “(1)Ineffective management, (2)Lack of or inadequate personnel training, (3) Lack of staff expertise and commitment and (4) Lack of knowledge about HACCP.
Authors must indicate in the method section what is the training and formation/education of people involved in the study. I agree with the author that 2nd, 3rd and 4th factors indicated above, are the main constrains in the implementation of a HACCP system in a food industry, mainly in those related of food of animal origin (meat and fish mainly). I have several years of experience as food safety consultant in meat industry and the main problem, not discussed by authors, is related to the training of the food safety/quality staff. Most of them are mainly BSc or MSc food engineers that do not have enough knowledge about slaughter processes, biological and chemical hazards of livestock (including zoonoses) and meat microbiology. Unfortunately, scarce veterinarians are employed in meat industries to guarantee the food safety/quality departments. Even those researches with PhD position indicated in table 3, must indicate if expertise is really related with food safety.
2.2. Line 336. Authors indicated “complicated guidelines”. Although I am totally agree with authors, not discussion is provided. The paper “García-Díez, J., Moura, D., Nascimento, M. M., & Saraiva, C. (2018). Performance assessment of open-access information about food safety. Journal of Consumer Protection and Food Safety, 13, 113-124.” discuss the deficiencies of sectorial and official guidelines about HACCP implementation.
Author Response
We thank for your interest in our work and for the valuable and helpful comments that will greatly improve the manuscript. We have tried to do our best to respond to the points raised in a timely manner. As indicated below, we have checked all the comments provided and have made necessary changes accordingly.
All revisions made to the manuscript have been marked up using red color.
General comments
The presented article utilizes the DEMATEL technique to assess the main problems in the implementation of an HACCP plan in the food industry. The results are consistent with the reality of the companies. The primary issue with the methodology is the limited information collected about the companies. I consider it important to include the volume (in tons) per year, as well as the number of products they produce. It is also not indicated whether all companies have one (or several) individuals dedicated to ensuring the HACCP plan or if it is an external company (outsourcing) that carries it out. Furthermore, it is not specified whether the training is specific to food safety, only indicating whether it is professional, university, or doctoral education.
Lastly (and associated with the scant description of both the characteristics of the sample individuals and the company's characterization), the discussion is scarce or almost non-existent. I find the work interesting but it does not discuss the reasons behind the results.
Thank you for your insightful comments and constructive feedback on our article. We are pleased that you found the results consistent with industry realities.
Regarding the methodology, we acknowledge the importance of providing more detailed information about the characteristics of the companies included in the study. We have addressed these concerns in the text, specifically in lines 302-310, where we stated that all organizations within the meat production industry included in the study have a minimum operational history of 10 years and produce a minimum of 3 distinct products. Furthermore, we clarified that quantifying volume in tons per year was deemed impractical and irrelevant for meaningful comparison due to the variability in meat types and production processes. Additionally, we highlighted that all companies in the sample maintain internal personnel dedicated to HACCP plan oversight, with no instances of external outsourcing for this function. Respondents' professional qualification levels correspond to their nationally recognized educational diplomas, although the majority reported participation in short-term training programs focused specifically on food safety.
Furthermore, we agree with your observation about the discussion section's brevity and recognize the need to highlight the reasons behind the results. We have provided additional insights and discussion on these matters in lines 442-454.
Specific comments
- Introduction
1.1. Line 55: HACCP's systematic approach, particularly in primary production, plays a pivotal role in preventing microbiological contamination and enhancing food quality and certification under HACCP is perceived as a crucial tool for producers to signal product quality and safety to consumers.
The HACCP in the primary production is not compulsory in the EU. If authors observe the 7 points of the haccp implementation, the 4th point indicates the “need of monitoring procedures”. For example, in a dairy farm, you can identify Brucella, Mycobacterium spp., Coxiella spp., Chlamydia spp. etc as a microbiological risk. After that, you can monitorize the microbiological counts in the bulk tank (independently its economic viability). But then, how do you apply corrective measures? Now, imagine you produce lettuces, How do authors apply a “monitoring system”? After that explanation, the HACCP does not play a pivotal role in the microbiological contamination in the primary production as stated. It´s impossoble to implement a HACCP program in the primary production.
We appreciate the your insightful critique regarding the terms 'particularly in primary production' from line 55 of our article. We have eliminated this part to accurately reflect the challenges associated with implementing HACCP in primary production settings, as highlighted in the review.
- Metodology – results
2.1. In the results sections, author stated that the first 4 factors (line 320-325) “(1)Ineffective management, (2)Lack of or inadequate personnel training, (3) Lack of staff expertise and commitment and (4) Lack of knowledge about HACCP.
Authors must indicate in the method section what is the training and formation/education of people involved in the study. I agree with the author that 2nd, 3rd and 4th factors indicated above, are the main constrains in the implementation of a HACCP system in a food industry, mainly in those related of food of animal origin (meat and fish mainly). I have several years of experience as food safety consultant in meat industry and the main problem, not discussed by authors, is related to the training of the food safety/quality staff. Most of them are mainly BSc or MSc food engineers that do not have enough knowledge about slaughter processes, biological and chemical hazards of livestock (including zoonoses) and meat microbiology. Unfortunately, scarce veterinarians are employed in meat industries to guarantee the food safety/quality departments. Even those researches with PhD position indicated in table 3, must indicate if expertise is really related with food safety.
We appreciate the your valuable insights regarding the importance of detailing the training and education of personnel in the methodology section. In response, we have included additional information in the Results and Discussion section (lines 302-310 and 343-351) to address this concern. Our study acknowledges the significance of personnel qualifications and expertise, highlighting the need for comprehensive training programs to address gaps in knowledge, particularly in areas such as slaughter processes and meat microbiology.
2.2. Line 336. Authors indicated “complicated guidelines”. Although I am totally agree with authors, not discussion is provided. The paper “García-Díez, J., Moura, D., Nascimento, M. M., & Saraiva, C. (2018). Performance assessment of open-access information about food safety. Journal of Consumer Protection and Food Safety, 13, 113-124.” discuss the deficiencies of sectorial and official guidelines about HACCP implementation.
We acknowledge your observation and have addressed it by incorporating the citation provided (lines 374-380). The added discussion elaborates on the challenges posed by complicated guidelines in HACCP implementation, drawing from Garcia-Díez et al. (2018) to support our argument. This citation enhances the paper's discussion by providing additional context and supporting evidence regarding the deficiencies of sectorial and official guidelines in ensuring effective HACCP implementation and food safety compliance.
Once again, we sincerely appreciate your thorough review and constructive feedback, which have undoubtedly strengthened the quality of our manuscript.
Reviewer 2 Report
Comments and Suggestions for Authors
The article "Exploring Key Barriers of HACCP Certification Adoption in the Meat Industry: A Decision-Making Trial and Evaluation Laboratory Approach" has an interesting objective, but its structure is questionable - is it a research article or a review article?
Considering that intended to be a research article, a more objective structure in the introduction is relevant (it may not be adapted to a literature review), but much of what we find in the introduction and even the results (Table 3) should be in the materials and methods. On the other hand, the results (which are too exhaustive) don't present any discussion, which appears lightly in the conclusion. So, if it's a research article, a major change in its structure is essential.
If it's a review article, it should be specific to the topic (for example, it discusses the methodology used in the fish industry) and we wonder how it adds to the evaluation carried out using this methodology.
A thorough structural revision and clarification of the article is suggested.
Comments on the Quality of English Language
Correct and easy to read scientific language
Author Response
We thank for your interest in our work and for the valuable and helpful comments that will greatly improve the manuscript. We have tried to do our best to respond to the points raised in a timely manner. As indicated below, we have checked all the comments provided and have made necessary changes accordingly.
All revisions made to the manuscript have been marked up using red color.
The article "Exploring Key Barriers of HACCP Certification Adoption in the Meat Industry: A Decision-Making Trial and Evaluation Laboratory Approach" has an interesting objective, but its structure is questionable - is it a research article or a review article?
Considering that intended to be a research article, a more objective structure in the introduction is relevant (it may not be adapted to a literature review), but much of what we find in the introduction and even the results (Table 3) should be in the materials and methods. On the other hand, the results (which are too exhaustive) don't present any discussion, which appears lightly in the conclusion. So, if it's a research article, a major change in its structure is essential.
If it's a review article, it should be specific to the topic (for example, it discusses the methodology used in the fish industry) and we wonder how it adds to the evaluation carried out using this methodology.
A thorough structural revision and clarification of the article is suggested.
Thank you for your thoughtful review and valuable feedback on our paper titled "Exploring Key Barriers of HACCP Certification Adoption in the Meat Industry: A Decision-Making Trial and Evaluation Laboratory Approach." We appreciate your insights regarding the structure and clarity of the article.
We acknowledge your concern regarding the structure of the paper and its alignment with either a research or review article format. We would like to clarify that our intention was to present a research article utilizing the DEMATEL technique, a quantitative research method, to explore key barriers to HACCP certification adoption in the meat industry. The items identified for analysis were based on a comprehensive review of the relevant literature.
In response to your feedback, we have carefully reconsidered the organization of the manuscript. We have made significant revisions to enhance clarity and ensure that the content aligns more effectively with the intended research focus. Specifically, we have expanded the Results and Discussion section to provide greater detail on the relevant findings and to foster a more cohesive narrative.
We understand the importance of maintaining a clear and objective structure in research articles and have endeavored to address your concerns through these revisions. We believe that these changes will contribute to a more coherent and impactful presentation of our research findings.
Once again, we sincerely appreciate your thorough review and constructive feedback, which have undoubtedly strengthened the quality of our manuscript.
Reviewer 3 Report
Comments and Suggestions for Authors
According to authors, ‘This study investigates an ongoing and significant concern in the meat industry, respectively the main barriers of HACCP implementation, being the first approach of the DEMATEL method in this context’. The subject is interesting, the review provided in the first section of the manuscript is helpful and the approach presented could be really useful and educational. Some minor points to be corrected:
Line 51: reflecting its recognition…state exactly the number of EU legislation that mandated HACCP
Lines 61-65: too long sentence, rephrase and use smaller sentences, for an easier comprehension
Line 123: In my opinion, the identification of the potential barriers is very important and should be included in the main part, not as an Appendix.
Line 175: barriers for HACCP
Line 185: what does FSMS stand for?
Line 227: You mention Table 3 before the first time you mention Table 2
Line 238: you mean that n=18 in your case? Explain explicitly the number of rows and columns of Table (line 239). Is it 12x18?
Line 272-273: the symbol here again is n=number of barriers? In line 238, you use again the same symbol (n) for the number of experts. Check again your nomenclature to avoid misunderstandings and confusion.
In Fig 1, check the significant digits in y-axis.
Can you also add a more comprehensive and simple diagram to depict clearly cause and effect (such as Pareto or fishbone)?
Comments on the Quality of English LanguageGood quality of English language, minor editing required.
Author Response
We thank for your interest in our work and for the valuable and helpful comments that will greatly improve the manuscript. We have tried to do our best to respond to the points raised in a timely manner. As indicated below, we have checked all the comments provided and have made necessary changes accordingly.
All revisions made to the manuscript have been marked up using red color.
According to authors, ‘This study investigates an ongoing and significant concern in the meat industry, respectively the main barriers of HACCP implementation, being the first approach of the DEMATEL method in this context’. The subject is interesting, the review provided in the first section of the manuscript is helpful and the approach presented could be really useful and educational. Some minor points to be corrected:
Line 51: reflecting its recognition…state exactly the number of EU legislation that mandated HACCP
Lines 61-65: too long sentence, rephrase and use smaller sentences, for an easier comprehension
Line 123: In my opinion, the identification of the potential barriers is very important and should be included in the main part, not as an Appendix.
Line 175: barriers for HACCP
Line 185: what does FSMS stand for?
Line 227: You mention Table 3 before the first time you mention Table 2
Line 238: you mean that n=18 in your case? Explain explicitly the number of rows and columns of Table (line 239). Is it 12x18?
Line 272-273: the symbol here again is n=number of barriers? In line 238, you use again the same symbol (n) for the number of experts. Check again your nomenclature to avoid misunderstandings and confusion.
In Fig 1, check the significant digits in y-axis.
Can you also add a more comprehensive and simple diagram to depict clearly cause and effect (such as Pareto or fishbone)?
We greatly appreciate your thorough review and constructive feedback on our manuscript titled "Exploring Key Barriers of HACCP Certification Adoption in the Meat Industry: A Decision-Making Trial and Evaluation Laboratory Approach." We have carefully addressed each of the points you raised:
Line 51: We have clarified the reference to EU legislation mandating HACCP implementation by adding the General Food Law Regulation (EC) No 178/2002.
Lines 61-65: We have revised the sentence structure to improve readability and comprehension.
Line 123: We acknowledge your perspective regarding the integration of all potential barriers into the main body rather than as an Appendix. Nevertheless, we have opted to maintain the extensive list of barriers in the Appendix and retain the concise list of factors included in the questionnaire within the main text. This decision was made to enhance the readability of the main text while ensuring that readers have access to supplementary details.
Line 175: We have corrected the reference.
Line 185: We have expanded "FSMS" to "Food Safety Management System" for clarity.
Line 227: This order was maintained because Table 2, "The DEMATEL scale equivalent value," is the first table presented in this section, followed by Table 3, "Sample of participating experts in the DEMATEL study." We have maintained the original order of table references in line with their appearance in the manuscript sections.
Line 238: We have provided explicit clarification regarding the dimensions of the matrix, specifying that it consists of 12 rows and 18 columns.
Line 272-273: We have ensured consistency in the use of symbols as "n" refers to the number of barriers in both instances.
Fig 1: We have adjusted the significant digits on the y-axis for clarity.
Regarding the recommendation for a more comprehensive diagram, we have opted to maintain the DEMATEL diagram (Figure 2), which aligns with the methodology utilized in our investigation.
Thank you once again for your insightful comments, which have contributed to the improvement of our manuscript.
Round 2
Reviewer 1 Report
Comments and Suggestions for Authors
Dear authors
As indicated in the previous review, I consider your work very interesting. The methodology authors described used is not usually used in the food safety context to evaluate barriers in the implementation of food safety plans.
Although authors improve some indications I made, the main problem indicate is still present: the manuscript described very well the results but there is no any discussion.
In table A1 in the appendix A, authors described a literature review of main constraints in the implementation of an HACCP program. So, I recommend authors to use this valuable information to make a proper discussion of the results.
Comments on the Quality of English LanguageMinor revision is required
Author Response
Dear Reviewer,
Thank you for your valuable feedback and for acknowledging the significance of our research. We appreciate your acknowledgment of the novelty and interest in our research methodology, particularly its application in the context of food safety.
We have carefully considered your suggestions and have made substantial revisions to address the discussion in our manuscript. In response to your recommendation, we have utilized the information from Table A1 in Appendix A, which presents a comprehensive literature review of the main constraints in the implementation of an HACCP program. We have incorporated valuable information into our discussion sections, specifically in the paragraphs highlighted in red within the text (lines 341-346, 348-357, 360-366, 368-369, 413-441, 477-481). By integrating these insights, we aim to provide a more robust analysis of our results and their implications within the broader context of food safety management.
While we acknowledge that the topic is vast and the discussion could be complex, we believe that the revised version of the manuscript now adequately addresses the essential aspects of our research findings and their implications. We are confident that the new version of the article effectively addresses your concerns and is suitable for publication.
Once again, we thank you for your valuable feedback and appreciate the opportunity to improve our work.
Reviewer 2 Report
Comments and Suggestions for Authors
Thank you for your comments and information. I still think the weakest point of the article is its structure, which has not been changed sufficiently. Here are some examples:
1. The journal advises that research manuscripts should include:
Front matter: Title, list of authors, affiliations, abstract, keywords.
Research manuscript sections: Introduction, Materials and Methods, Results, Discussion, Conclusions (optional).
Back matter: Supplementary Materials, Acknowledgements, Author Contributions, Conflicts of Interest, References.
I agree with this structure, but it's not the same as the article in question, especially in the introduction, with a literature review. I believe that the introduction should be revised and adapted to include the information relevant to the understanding of the article.
Similarly, much of the material and methods is in the results (lines 249-253; Table 3, lines 302-317, .....). Methods provide results, as we all know, and the results chapter should describe and discuss the results. However, the description of methods and results is preferred and little is discussed. It is only on line 366 that you really start to analyse and discuss the results.
Furthermore, because the discussion is short, the conclusions are reinforced, which I don't think is the best strategy. So, without compromising the scientific interest, I still think it is necessary to revise the structure of the article.
Comments on the Quality of English Language
Correct and easy to read
Author Response
Dear Reviewer,
Thank you for your thoughtful feedback and suggestions regarding the structure of our manuscript. We appreciate the opportunity to address your concerns and provide clarification on the changes made in response to your comments.
We acknowledge your observation regarding the structure of the paper and the importance of adhering to the standard format recommended by the journal. We have carefully reviewed your suggestions and have made several revisions to enhance the clarity and coherence of the manuscript.
In particular, we have incorporated the information from Table A1 in Appendix A into the Results and Discussion section, as highlighted in red between lines 341-346, 348-357, 360-366, 368-369, 413-441, and 477-481. This addition has allowed us to provide a more comprehensive analysis and interpretation of the results obtained, addressing the lack of discussion noted in the initial review.
Furthermore, we have moved two paragraphs from the Introduction to the Literature Review (lines 94-101) and Materials and Methods (lines 197-218), as highlighted in yellow, to improve the flow and organization of the paper. We have also revised the reference section to match the new modifications.
Regarding the structure of the paper, we would like to highlight that the current format aligns well with the conventions followed in the majority of published papers on DEMATEL topics in prestigious journals, such as the following ones:
- Chen, W. K., Nalluri, V., Hung, H. C., Chang, M. C., & Lin, C. T. (2021). Apply DEMATEL to analyzing key barriers to implementing the circular economy: an application for the textile sector. Applied Sciences, 11(8), 3335.
- Akal, A. Y., Kineber, A. F., & Mohandes, S. R. (2022). A phase-based roadmap for proliferating BIM within the construction sector using DEMATEL technique: perspectives from Egyptian practitioners. Buildings, 12(11), 1805.
- Khan, S., Singh, R., Haleem, A., Dsilva, J., & Ali, S. S. (2022). Exploration of critical success factors of logistics 4.0: a DEMATEL approach. Logistics, 6(1), 13.
- Zhao, G., Ahmed, R. I., Ahmad, N., Yan, C., & Usmani, M. S. (2021). Prioritizing critical success factors for sustainable energy sector in China: A DEMATEL approach. Energy Strategy Reviews, 35, 100635.
These examples demonstrate that our paper follows a similar structure to other relevant studies using DEMATEL as a research methodology.
We believe that the revisions made have addressed the structural issues raised and have improved the overall quality of the manuscript. We are confident that the new version of the article effectively addresses your concerns and is suitable for publication.
Once again, we thank you for your valuable feedback and appreciate the opportunity to improve our work.
Round 3
Reviewer 2 Report
Comments and Suggestions for Authors
I congratulate the authors on the changes they have made, which have made the article more understandable and in a more correct scientific structure.
In my opinion, it provides the minimum clarity needed for an article to be suitable for publication.